# Anisotropic long-range spin transport in canted antiferromagnetic orthoferrite YFeO₃

Shubhankar Das[1], A. Ross [2,8], X. X. Ma [3,8], S. Becker [1], C. Schmitt [1], F. van Duijn [4,5], E. F. Galindez-Ruales [1], F. Fuhrmann[1], M.-A. Syskaki [1], U. Ebels[4], V. Baltz [4], A.-L. Barra[5], H. Y. Chen[3], G. Jakob [1,6], S. X. Cao [3] ✉, J. Sinova[1], O. Gomonay [1], R. Lebrun [2] & M. Kläui[1,6,7] ✉

In antiferromagnets, the efficient transport of spin-waves has until now only been observed in the insulating antiferromagnet hematite, where circularly (or a superposition of pairs of linearly) polarized spin-waves diffuse over long distances. Here, we report long-distance spin-transport in the antiferromagnetic orthoferrite YFeO₃, where a different transport mechanism is enabled by the combined presence of the Dzyaloshinskii-Moriya interaction and externally applied fields. The magnon decay length is shown to exceed hundreds of nanometers, in line with resonance measurements that highlight the low magnetic damping. We observe a strong anisotropy in the magnon decay lengths that we can attribute to the role of the magnon group velocity in the transport of spin-waves in antiferromagnets. This unique mode of transport identified in YFeO₃ opens up the possibility of a large and technologically relevant class of materials, i.e., canted antiferromagnets, for long-distance spin transport.

The field of antiferromagnetic spintronics seeks to functionalize the high frequency spin dynamics, resilience to external magnetic fields and lack of stray fields of antiferromagnetic materials for information storage and transfer[1,2]. The emerging promising subfield of antiferromagnetic magnonics seeks to make use of insulating antiferromagnets to transport angular momentum for information processing via magnons, the quanta of magnetic excitation[3] where magnons can be excited and transported via both the Néel vector (**n**) and potential net magnetic moments (**m**)[4]. In easy-axis antiferromagnets, spin angular momentum can be transferred by the circularly polarized magnon eigenmodes, enabling for instance long-distance transport of magnons in the low temperature easy-axis phase of the antiferromagnetic iron oxide, hematite (α-Fe₂O₃)[5,6]. Easy-plane antiferromagnets on the other hand have linearly polarized magnon

eigenmodes, which conventionally do not carry angular momentum[7,8]. However, a superposition of modes that dephase has been shown to transport magnons efficiently in the room temperature, easy-plane phase of α-Fe₂O₃[9–11]. Although long-distance magnon transport has been shown in ferrimagnets[12,13], ferromagnets with antiparallelly coupled domains[14], multiferroic/ferromagnetic heterostructures[15] and both easy-axis and easy-plane α-Fe₂O₃[5,9], magnon transport exceeding decay-lengths of a few nanometers has yet to be demonstrated in any other antiferromagnet of either easy-axis or easy-plane anisotropy. This raises the question of whether there is something unique about hematite amongst the compendium of antiferromagnetic insulators or whether other antiferromagnets can exhibit long distance angular momentum transport and in particular what magnetic properties are required.

[1]Institute of Physics, Johannes Gutenberg University Mainz, Staudingerweg 7, 55128 Mainz, Germany. [2]Unité Mixte de Physique CNRS, Thales, Université Paris-Saclay, Palaiseau 91767, France. [3]Department of Physics, Materials Genome Institute, International Center for Quantum and Molecular Structures, Shanghai University, Shanghai 200444, China. [4]Univ. Grenoble Alpes, CNRS, CEA, Grenoble INP, SPINTEC, F-38000 Grenoble, France. [5]Laboratoire National des Champs Magnétiques Intenses, CNRS-UGA-UPS-INSA-EMFL, F-38042 Grenoble, France. [6]Graduate School of Excellence Materials Science in Mainz, Staudingerweg 9, 55128 Mainz, Germany. [7]Center for Quantum Spintronics, Norwegian University of Science and Technology, Trondheim 7491, Norway. [8]These authors contributed equally: A. Ross, X. X. Ma. ✉e-mail: sxcao@shu.edu.cn; klaeui@uni-mainz.de

Although α-Fe₂O₃ has shown to exhibit micrometer magnon decay lengths across a large temperature range, the transport efficiency drops rapidly across the easy-axis to easy-plane phase transition, the Morin transition[9]. Furthermore, the antiferromagnetic anisotropy in hematite is highly temperature sensitive[16], meaning that diffusing magnons experience regions of differing anisotropy in the presence of thermal gradients, limiting the potential for active manipulation of the magnon current by, for example, localized Joule heating[17]. It has been demonstrated that dilute doped hematite with a non-magnetic ion still facilitates efficient magnon transport despite the changes in magnetic symmetry that occur[18]. If this doping is instead taken to the extreme, where fifty percent of the Fe is substituted, a large class of materials, known as orthoferrites, appear and these are expressed with the formula XFeO₃. These materials have a range of interesting properties that can be explored in conjunction with magnon transport such as piezo-electricity[19–21] and a large magnetostriction[22,23]. For the most part, this class of materials are antiferromagnetic with a range of symmetries depending on the exact element X[24]. Due to Dzyaloshinskii-Moriya interaction (DMI), ortho-ferrites also typically have a net magnetization that arises due to canting of the magnetic sublattices, that appears perpendicular to the DMI vector, if the DMI is not parallel to the Néel vector[25,26]. Epitaxial thin films of YFeO₃ show a ferroelectric transition at room temperature, while preserving its magnetization, due to the presence of Y-Fe anti-site defects that facilitate a non-centrosymmetric distortion which induces a spontaneous polarization[20]. Also, a spontaneous spin reorientation occurs in some orthoferrites where the rare earth atoms carry non-zero magnetic moments, such as TmFeO₃ and ErFeO₃[27]. Furthermore, this large class of materials can also exhibit potentially low damping, one of the key requirements for long distance spin transport. Thus, this materials class offers an exciting playground for probing the transport mechanisms of antiferromagnetic materials through a range of symmetries and anisotropies to unravel the critical criteria for efficient magnon transport and control.

Here we demonstrate magnon transport over micro-meters in the ultra-low damping, orthoferrite YFeO₃ making use of a transport mode unique to this type of system. This key observation of a promising transport mode opens up a large number of technologically relevant materials for low power, antiferromagnetic magnonic devices. Using an external magnetic field to manipulate the magnon eigenmode polarization, we characterize the transport efficiency and identify the dominant role of the Néel vector. Although the zero-field magnon eigenmodes are non-degenerate, unlike in easy-axis hematite, they are linearly polarized and unable to facilitate zero-field magnon transport. On applying an external field, the magnon eigenmodes become elliptically polarized when the field has a nonzero projection on the Néel vector, thus enabling efficient magnonic spin current transport over long distances. Both the exponential decay of spin transport signal with distance and its vanishing magnitude at lower temperature indicate a diffusive nature to the transport. The transport is furthermore anisotropic and we demonstrate that the magnon decay length varies between the different crystallographic axes, which can be explained by the anisotropic magnon group velocity arising from anisotropic exchange stiffness.

Antiferromagnetic YFeO₃ ($T_N = 644\,K$)[24,28] crystallizes in an orthorhombic structure[29] and has three principal anisotropy axes. The [100] a-axis and [010] b-axis are the antiferromagnetic easy and hard axes respectively, whilst the [001] c-axis is an intermediate anisotropy axis (see Fig. 1a). A strong DMI of 14 T parallel to the b-axis leads to a net magnetic moment orientated along the c-axis[30,31]. An external magnetic field applied along the a-axis initiates a smooth rotation of **n** confined within the ac-plane until **n** aligns parallel to the c-axis at the critical field ($H_{cr}$). The value of $\mu_0 H_{cr}$ varies between 7.4 T and 6.5 T at temperatures between 4.2 and 300 K[32,33]. Such a rotation under an applied field is caused by the homogeneous DMI[25], and

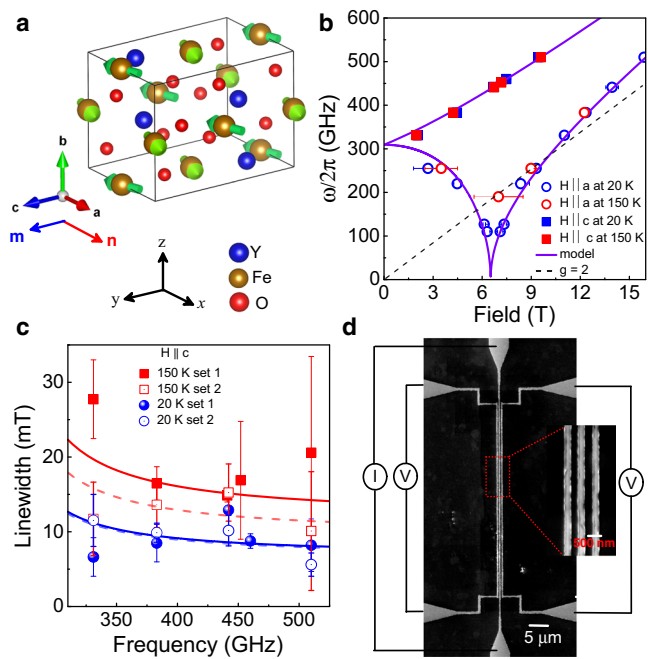

**Fig. 1 | Crystal structure, magnetic resonance and the device. a** Crystal structures of YFeO₃[010], where the nearest neighbour Fe-ions are coupled antiferromagnetically along the easy-axis ([100] a-axis) and the moments are slightly canted along the c-axis which yield a net weak magnetization directed parallel to the c-axis. The arrows in the structure indicate the direction of spins of the Fe ions. **b** Resonance frequency as a function of magnetic field along a-axis (open symbol) and c-axis (closed symbol) for two different temperatures, 20 K in blue and 150 K in red. The model parameters for the fits are $\mu_0 H_E = 635T$, $\mu_0 H_a = 0.19T$, $\mu_0 H_b = 0.7T$ and $\mu_0 H_{DMI} = 12T$. The g = 2 line shows the frequency of a potential spurious paramagnetic resonance. **c** Linewidth as a function of frequency for the configuration of H along c-axis at 20 K and 150 K. The open and closed symbols correspond to data taken separately, after removal and reintroduction of the sample, to test reproducibility. The solid solid and dashed lines are the theoretical fitting using the model from Fink[42], which yields damping coefficients of $3.5 \pm 0.4 \times 10^{-6}$ and $3.4 \pm 0.3 \times 10^{-6}$ for the two sets of data at 20 K and $6.2 \pm 0.3 \times 10^{-6}$ and $4.6 \pm 0.5 \times 10^{-6}$ for 150 K. The extracted parameters from (**b**), $\mu_0 H_E$, $\mu_0 H_a$, $\mu_0 H_b$ and $\mu_0 H_{DMI}$ are used in this model. **d** SEM image of a typical device, where the charge current is driven along the middle wire and the non-local voltages are measured in both wires to the left and right of it. We associate orthogonal coordinates with the crystallographic axes: **x ∥ a**, **y ∥ c**, **z ∥ b**.

contrasts with the abrupt spin-flop transition typical in purely collinear antiferromagnets. The DMI found in YFeO₃ is significantly stronger than that found in bulk hematite[34] and correspondingly the net moment is larger (see Supplementary Fig. 4). This actually enables the possibility to disentangle the feasibility for antiferromagnetic transport dictated by the net magnetic moment. Also, coherent control of precession motion of magnetizations has also been demonstrated with double pulse terahertz waves[35]. Although, due to the centro-symmetric orthorhombic structure, ferroelectricity is conceptually prohibited in YFeO₃, an unexpected weak ferroelectric polarization is observed in the bulk sample[36], whereas epitaxial thin films show much stronger ferroelectric polarization due to a Y-Fe antisite defects[20]. Also epitaxial YFeO₃ thin films deposited by adopting a hexagonal symmetry exhibit ferroelectricity[21]. Unlike hematite, this material has no temperature-driven spin transition[37] and the antiferromagnetic anisotropy is approximately constant over a broad temperature range making it potentially more suitable for device applications. It has also been shown to exhibit fast domain wall motion[38–40] that offers the potential for magnon driven domain wall motion for information storage.

## Results

### Magnetic damping in YFeO₃

Before exploring non-local spin transport experiments, we first determine if the magnetic Gilbert damping coefficient ($\alpha_G$), a key parameter for determining the spin-transport length scale, is low enough. To acquire information about the magnetization dynamics and $\alpha_G$, we characterize the magnetic resonance of a 0.5 mm thick [010]-oriented single crystal of YFeO₃ using a continuous-wave electron paramagnetic resonance spectrometer operating at high frequencies from 127 GHz to 510 GHz[9,41]. The antiferromagnetic resonance measurements show multiple peaks associated with magnetostatic modes (shown in Supplementary Figs. 6–9). Figure 1b shows frequency vs. resonance field for the low frequency magnon mode for a field applied along both the c-axis and a-axis at both temperatures 20 K and 150 K. The frequency dependence is fitted using the model described below and in Ref. 9. The extracted parameters from the fitting are $\mu_0 H_E = 635\,T$, $\mu_0 H_a = 0.19\,T$, $\mu_0 H_b = 0.7\,T$ and $\mu_0 H_{DMI} = 12\,T$, where $H_E$ is the exchange field, $H_a$ and $H_b$ are anisotropy fields associated with the easy (a-axis) and hard (b-axis) magnetic direction, respectively, $H_{DMI}$ is the homogeneous DMI field. We extract the average linewidth from the region with small peak-to peak distances of the resonances as indicated in Supplementary Figs. 6–9, which relate to the damping coefficient. The error bars are determined by the maximum deviation of the averaged value. Figure 1c and Supplementary Fig. 5 show the frequency dependence of the average linewidth for the low frequency magnon eigenmode for the field directed along the c-axis and a-axis, respectively. The linewidth dependence is fitted using the theoretical model of the antiferromagnetic resonance provided by Fink[42] (see Fig. 1c and Supplementary Fig. 5). This model reproduces the gradual decrease of the linewidth with frequency determined experimentally, taking into account the error bars. The fits returned $\alpha_G$ of $3.5 \pm 0.4 \times 10^{-6}$ and $6.2 \pm 0.3 \times 10^{-6}$ at 20 K and 150 K, respectively, for field along c-axis and $6 \pm 2 \times 10^{-6}$ at 20 K for field along a-axis. For H ∥ c, resonance data were measured several times, after removal and repositioning of the sample to test data reproducibility at both temperature at 20 K and 150 K, shown in Supplementary Figs. 6 and 7, respectively. Fitting of these sets of data (shown in Fig. 1c) yields $\alpha_G$ of $3.4 \pm 0.3 \times 10^{-6}$ and $4.6 \pm 0.5 \times 10^{-6}$ at 20 K and 150 K, respectively. Hence, the magnetic damping of YFeO₃ is of the same order of YIG[43] (the ferrimagnetic material with lowest reported $\alpha_G$), hematite[9] (the only antiferromagnetic material which has previously shown long distance spin transport) and lower than the metallic ferromagnet $Co_{25}Fe_{75}$[44] and $La_{0.7}Sr_{0.3}MnO_3$[45]. We note that the damping constant for an antiferromagnetic material inherently accounts for the strong internal exchange coupling between the two sub-lattices[46]. This field is 635 T for YFeO₃, which can explain the low magnitude of the damping. We also note that the data we obtain is also in agreement with the expected temperature dependence, i.e., a larger damping value for higher temperatures, related to the increase of magnonic and phononic contributions that open angular momentum relaxation channels. Finally, we would like to mention that we have been particularly careful not to underestimate the damping value and that our analysis provides an upper bound estimate[9].

### Devices and detection technique

To investigate the spin transport in [010]-oriented YFeO₃ (see methods and the supplementary for growth and characterization details), we make use of devices (see Fig. 1d) with two different geometries; where Pt wires are parallel and perpendicular to the easy-axis. When a charge current ($I$) is driven through the injector Pt wire, a transverse spin current is generated due to the spin Hall effect (SHE), which yields a spin accumulation $\mu_s$ at the Pt/YFeO₃ interface, polarized perpendicular to the flow direction of the charge current[47,48]. The spin current is absorbed by the antiferromagnet when $\mu_s$ is parallel to $\mathbf{n}$, resulting in a nonequilibrium distribution of magnons with an average spin parallel

to $\mu_s$. These magnons then diffuse in the YFeO₃, away from the injector. The magnon current flowing in the YFeO₃ is absorbed by a spatially and electrically separated Pt detector, where it is converted to a measurable voltage $V_{el}$ by the inverse-SHE[5,12]. As $V_{el}$ reverses its sign on reversing the current polarity, owing to the SHE[49] that reverses the spin current direction, we calculate the $V_{el}$ signal as ($V_{el}$ (I+)−$V_{el}$ (I-))/2 to remove polarity-independent effects[5,50]. We then express the spin transport signal as the nonlocal resistance $R_{el} = V_{el}/I$ (in units of Ohm).

### Spin transport for wires along the easy-axis

We first consider the spin transport in devices where the Pt-wires are parallel to the easy-axis (Fig. 2a). This geometry places the interfacial spin polarization $\mu_s$ perpendicular to $\mathbf{n}$ in the absence of a magnetic field (**H**). On sweeping the field along the easy-axis, $\mathbf{n}$ smoothly rotates due to the DMI within the ac-plane[25,26], reaching the c-axis (which is perpendicular to field) at $\mu_0 H_{cr}$ (=6.5 T) where it remains for $H > H_{cr}$. From the previous reports on hematite we expect that with the increased projection of $\mathbf{n}$ on $\mu_s$ with increasing field along the easy-axis, the $R_{el}$ signal should correspondingly increase and reach a maximum at $H_{cr}$[5,6]. As we increase the magnetic field applied to the YFeO₃ crystal, we see a gradual increase of $R_{el}$ as $\mathbf{n}$ rotates, however, contrary to hematite we observe a maximum below $H_{cr}$, followed by a sharp, sudden decrease before the signal vanishes at $H_{cr}$ and remains constant around zero within the error bar (shown in Fig. 2a). Therefore, the spin transport signal appears in the intermediate field value between $\mathbf{n}$ parallel and perpendicular to field. If instead, the field is applied perpendicular to the easy-axis, parallel to the $\mathbf{m}$ along the c-axis, no $R_{el}$ signal is observed across the whole field range (shown in Supplementary Fig. 11), which can be understood from the fact that $\mathbf{n}$ always remains perpendicular to both field and transport direction unlike in hematite. Hence, the response of the spin-transport signal by applying field along and perpendicular to easy-axis indicates that transport of spin information is mediated only by the Néel order. Any transport mediated by the net magnetic moment is negligible and remains within the experimental error. Thus, the transport is purely antiferromagnetic in nature. The solid line in Fig. 2a is the fit using theoretical modelling discussed later and in the supplementary, considering the spin angular momentum only mediated by the Néel vector, showing excellent agreement.

### Spin transport for wires perpendicular to the easy-axis

Having shown that the Néel vector facilitates efficient magnon transport along the intermediate anisotropy c-axis under an applied magnetic field, we next turn to investigating the spin transport along a different crystallographic axis, i.e., a-axis (the easy-axis) in order to probe the role of the anisotropy. To do this, we choose device where wires are perpendicular to the easy-axis. In such a configuration, at zero field, $\mathbf{n}$ is parallel to $\mu_s$. Previous reports on hematite show zero-field spin transport in such a configuration, as the magnon eigenmodes are circularly polarized at zero field and $\mathbf{n}$ has a finite projection on $\mu_s$[5,6]. However, in stark contrast, we find that in YFeO₃ there is no significant zero-field spin transport. On increasing the field along the easy-axis, the $R_{el}$ signal increases and reaches a maximum at $\mu_0 H = 3\,T$ followed by a decrease and the signal vanishes at $H_{cr}$ (shown in Fig. 2b). The mechanism for the increase in $R_{el}$ to a maximum and the decrease with increasing field can be explored by the equilibrium orientation of the $\mathbf{n}$, as described in previous geometry (shown in Fig. 2a). In this device geometry, we observe that the peak appears at lower fields than for the previous geometry as the spin-transport signal depends on two factors; the field dependence of the magnon magnetization and the projection of $\mathbf{n}$ on $\mu_s$. This behaviour is in line with the theoretical expectations as described in the section on the theoretical model below. Furthermore, on applying the field perpendicular to the easy-axis, no spin-transport signal is observed (shown in Supplementary Fig. 12), which indicates that the magnetic field perpendicular to $\mathbf{n}$ is

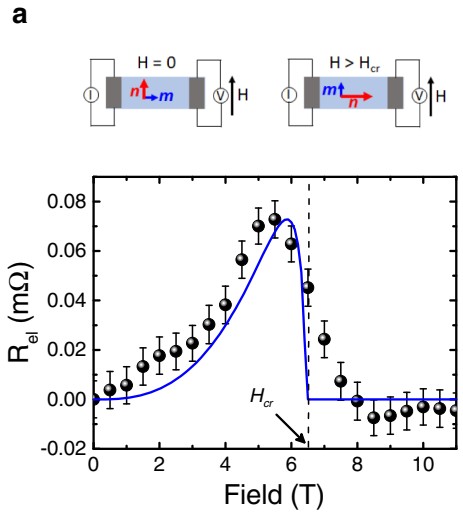

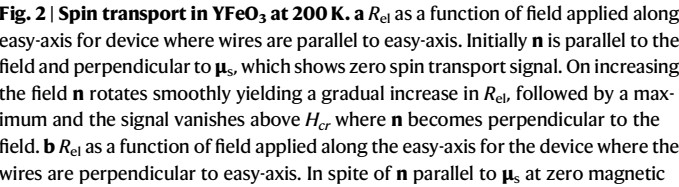

**Fig. 2 | Spin transport in YFeO₃ at 200 K. a** $R_{el}$ as a function of field applied along easy-axis for device where wires are parallel to easy-axis. Initially **n** is parallel to the field and perpendicular to **μ**ₛ, which shows zero spin transport signal. On increasing the field **n** rotates smoothly yielding a gradual increase in $R_{el}$, followed by a maximum and the signal vanishes above $H_{cr}$ where **n** becomes perpendicular to the field. **b** $R_{el}$ as a function of field applied along the easy-axis for the device where the wires are perpendicular to easy-axis. In spite of **n** parallel to **μ**ₛ at zero magnetic field, no zero-field spin transport is observed. $R_{el}$ increases with field initially and reaches a maximum at $\mu_0 H = 3$ T followed by a decrease and the signal diminishes above critical field. The error bars are the standard error of the mean. In both curves, the signal is plotted after subtracting the offset at zero field. The solid blue lines based on the theoretical model are described in the text. In both of the geometries, the centre-to-centre distance between the wires is about 525 nm.

unable to generate the required ellipticity to the magnon eigenmodes to enable magnon transport.

We also probed the field dependence of thermal magnon transport for both device geometries (shown in Supplementary Fig. 10) and establish that the transport of thermally-excited magnons is also mediated dominantly by Néel vector below $H_{cr}$, which is the sharp contrast to hematite where thermal spin transport is mainly mediated by field induced magnetic moment[5] (details are discussed in the supplementary).

Thus, as a first key result we can conclude that the angular momentum transport in YFeO₃ is significantly different from the previously observed transport in hematite with major qualitative differences. In particular, the dominance of the spin transport on the Néel vector orientation and magnon ellipticity is further confirmed by additional measurements varying the field direction with respect to the anisotropy axes (see Supplementary Fig. 13).

### Estimation of magnon decay length

To check if the transport mode present in YFeO₃ also lends itself to long distance transport, we measure the magnon decay length ($\lambda$) that reflects the efficiency of the spin transport. With the exception of hematite, all reported antiferromagnets have so far yielded $\lambda < -10$ nm[51–54]. To determine $\lambda$, we have measured $R_{el}$ as a function of centre-to-centre distance between the wires up to 1.25 μm for both of the geometries as shown in Fig. 3. $R_{el}$ is recorded at the maximum signal for each geometry i.e., at 5.5 T for devices shown in Fig. 2a and at 3 T for devices shown in Fig. 2b. Furthermore, to compare $\lambda$ along two anisotropy axes at a constant field, in order to rule out a possible field dependent reduction of $\lambda$ such as observed in ferromagnetic materials[55], $R_{el}$ is also recorded at 3 T for the former geometry. The curves are fitted with an exponential decay function: $R_{el} = Ae^{-d/\lambda}$, where $A$ is a distance independent pre-factor, and yield $\lambda = 470 \pm 40$ nm and $525 \pm 50$ nm at 5.5 T and 3 T, respectively, for the former device and $280 \pm 20$ nm at 3 T for the latter device. The exponential fitting indicates a diffusive nature of the transport[5,6,12] and excludes the possibility of dominating spin superfluidity[56–58]. Also, the temperature dependence of $R_{el}$ (shown in the supplementary Fig. 14) at low temperature points to the diffusive nature of the transport[9].

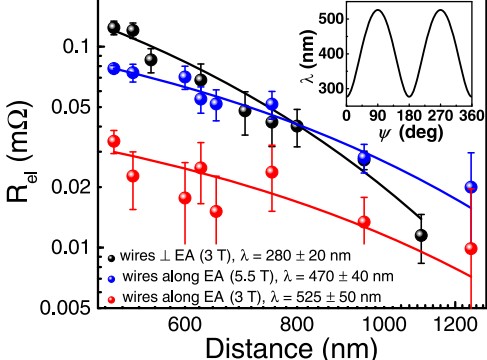

**Fig. 3 | Distance dependence of spin transport signal at 200 K.** Nonlocal spin transport signal $R_{el}$ as a function of centre-to-centre distance between of the wires for devices where wires are along the easy-axis at $\mu_0 H = 5.5$ T and 3 T and for devices where wires are perpendicular to the easy-axis at $\mu_0 H = 3$ T. The error bars are standard errors of the mean. The solid lines are the fit to an exponential decay function which yields value of $\lambda$ of $470 \pm 40$ nm and $525 \pm 50$ nm at 5.5 T and 3 T, respectively, for the devices shown in Fig. 2a and $280 \pm 20$ nm at 3 T for the devices shown in Fig. 2b. Inset, theoretical calculation of $\lambda$ as a function of angle $\psi$ measured from the easy-axis, considering the experimental value of $\lambda$ along and perpendicular to the easy-axis.

We observe a significant difference in magnon decay length scale along different anisotropy axes, where $\lambda$ is -50% lower along the $a$-axis than the $c$-axis. For ferromagnets, such an effect has been explained by anisotropy in the magnetic energy under the influence of magnetic dipole-dipole interaction[59]. In the case of antiferromagnets and weak ferromagnets dipole-dipole interactions are negligible. Hence, the anisotropy in the magnetic energy does not affect the group velocity of magnons, as is the case in ferro- or ferrimagnets. The observed anisotropy of $\lambda$ in our case can be explained by the anisotropy of the magnetic damping (or relaxation time) or anisotropy of the magnon group velocity along different anisotropy axes. Using the theoretical analysis discussed in detail below, we identify the anisotropy in the magnon group velocity as the main

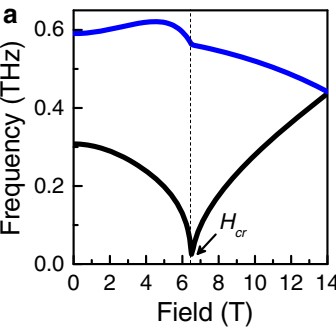
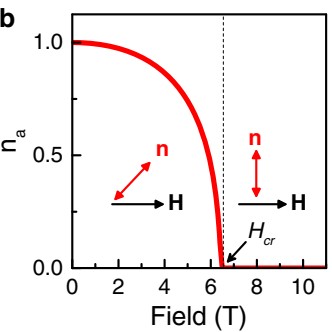
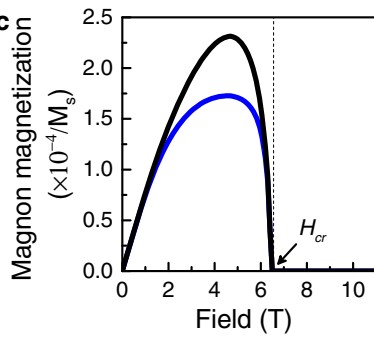

**Fig. 4 | Characteristics of two magnon eigenmodes for a magnetic field applied along the a-axis. a** Field dependence of the eigen-frequencies for two magnon eigenmodes. Softening of the low frequency eigenmode takes place for $H = H_{cr}$. **b** Component of the Néel vector along $a$-axis as a function of field applied along easy-axis, indicating a smooth orientation of Néel vector towards the $c$-axis. The critical field is defined at a field where $n_a$ is zero. **c** Integral contribution of all magnons into magnetization averaged over the **k**-space with the Bose distribution function for low-frequency and high-frequency branches. For fields along the easy-axis, the ellipticity is zero above $H_{cr}$, where **n** ⊥ **H** and the magnon modes are linearly polarized.

mechanism for the observed anisotropic magnon decay length (see inset in Fig. 3).

## Discussion

### Theoretical model of magnon spin transport in YFeO₃

To understand and interpret the magnon transport in orthoferrites, we first calculate the magnon eigenmodes by solving standard equations for antiferromagnetic dynamics in the presence of an external magnetic field (details in the supplementary). We then analyze the angular momenta of each mode, associating them with the dynamic magnetization $\mathbf{m}_{dyn} = M_s \delta \mathbf{n} \times \delta \dot{\mathbf{n}} / \gamma H_{ex}$.

The magnon spectra include two branches with frequencies,

$$\omega_{1,2} = \sqrt{\left(\omega_{1,2}^0(H)\right)^2 + c_a^2 k_x^2 + c_b^2 k_y^2 + c_c^2 k_z^2}, \quad (1)$$

where the $\omega_{1,2}^0(H)$ are field-dependent frequencies of antiferromagnetic resonance (see Fig. 4a), $c_a \neq c_b \neq c_c$ are limiting magnon velocities along different crystallographic directions ($a$, $b$, and $c$), and **k** is the magnon wave-vector. $\omega_{1,2}^0(H)$ depends on the material parameters, described in the supplementary. In the absence of an external field, all the eigenmodes are linearly polarized and correspond to oscillations of the Néel vector along either the $c$-axis (low-frequency eigenmodes $\omega_1(\mathbf{k})$) or along the $b$-axis (high-frequency eigenmodes $\omega_2(\mathbf{k})$). The magnetization of such modes oscillates with the magnon frequency and cannot transport time-independent (dc) angular momentum, yielding a zero-transport signal at zero field, as experimentally observed (and in contrast to hematite where zero-field transport is present). The linear polarization of the eigenmodes is a direct consequence of the strong orthorhombicity in which case $\omega_1^0 \neq \omega_2^0$ due to the different magnetic anisotropies along the $a$- and $b$-axes. It should be noted that the magnon velocity is also anisotropic and depends on the direction of **k**. However, this is related with anisotropy of the exchange stiffness (different exchange coupling along different axes[60], where $J_c > J_{ab}$, so, we can assume that $c_c > c_a$).

The external magnetic field **H**||**a** applied along the easy-axis induces a smooth reorientation of the Néel vector, confined to the $ac$-plane. Figure 4b shows the projection of the **n** along the $a$-axis as a function of the field along $a$-axis, where the projection of $H$ on the $b$-axis is zero. In this geometry, the magnetic field not only affects the values of $\omega_{1,2}^0(H)$, it also modifies the polarization of the eigenmodes from linearly polarized to elliptically polarized. The magnetization of the elliptically polarized modes also has a dc component, which is parallel to equilibrium orientation of the Néel vector, $\mathbf{m}_{dyn} \parallel \mathbf{n}^{(0)}$, and whose value is proportional to the product of magnon density, magnon frequency and magnon ellipticity (see Fig. 4c). All three contributions depend on the magnetic field. Consequently, an increase of the spin-transport signal with field is observed. The direction of the magnetization in the low-frequency magnon branch corresponds to $\mathbf{m}_{dyn} \cdot \mathbf{H} > 0$ and is opposite for the high-frequency branch. At the critical field $H_{cr}$, corresponding to the alignment of the Néel vector along the $c$-axis (perpendicular to the magnetic field), the ellipticity of the modes goes to zero and above $H \geq H_{cr}$ magnons are again linearly polarized and unable to transfer dc angular momentum.

The electrical, $R_{el}$, (and thermal, $R_{th}$, in the supplementary) transport signals are proportional to the weighted sum of the dc magnetizations of all available magnon eigenmodes. In addition, $R_{el} \propto \left(\mathbf{n}^{(0)} \cdot \hat{\mu}\right)^2$ which corresponds to the projection of the $\mathbf{m}_{dyn}$ on the direction of the spin accumulation $\hat{\mu} \parallel \boldsymbol{\mu}_s$ ($|\hat{\mu}| = 1$) during the pumping and detection of non-equilibrium magnons[9]. $R_{th} \propto \mathbf{n}^{(0)} \cdot \hat{\mu}$, as the magnetization is projected only during the detection. Assuming that contribution of different magnon modes is proportional to the thermal probability given by Bose distribution, we calculate the expected $R_{el}$, signal which we show as solid line fits to the experimental data in Fig. 2.

Next, we consider the diffusion of magnons that is responsible for the transport of angular momentum. The spin diffusion coefficient $D$ is calculated in the framework of linear nonequilibrium thermodynamics as a response to the gradient of spin accumulation $\boldsymbol{\mu}_s$ (see Ref. 4). The corresponding expression reads as:

$$D = \frac{1}{T} \int \frac{d^3k}{(2\pi)^3} \sum_{\alpha=1,2} \tau_\alpha(\mathbf{k}) \left(\mathbf{v}_{x\alpha}\cos\psi + \mathbf{v}_{y\alpha}\sin\psi\right)^2 \frac{(\mathbf{m}_\alpha\hat{\mu})\exp(\hbar\omega_\alpha/T)}{(\exp(\hbar\omega_\alpha/T) - 1)^2} \quad (2)$$

where $\tau_\alpha(\mathbf{k})$ and $\mathbf{v}_\alpha \equiv \partial\omega_\alpha/\partial\mathbf{k}$ are the relaxation time and group velocity of the magnon of the $\alpha$-mode with the wave vector **k**, respectively. The angle $\psi$, calculated from the $a$-axis, defines the direction of the spin accumulation gradient and $T$ is the temperature. Using the dispersions given in Eq. (1) and assuming that the relaxation time is the same for all magnon eigenmodes, we determine that the value of the diffusion coefficient $D$ is anisotropic, $D \propto c_a^2\cos^2\psi + c_c^2\sin^2\psi$. The inset of Fig. 3 shows the angular dependence of $\lambda \propto \sqrt{D} \propto \sqrt{c_a^2\cos^2\psi + c_c^2\sin^2\psi}$ calculated based on the assumption that $c_c \approx 38\text{km/sec} > c_a \approx 20$ km/sec, which is based on experimental observations[61]. From this, we conclude that the anisotropy observed in the magnon-decay length originates mainly from anisotropy of the exchange stiffness (or, at the microscopic level, from anisotropy of the exchange interactions in the $ab$-plane and along the $c$-axis[60]), though the anisotropy of the relaxation time can also contribute to the observed effect. This mechanism contrasts with the mechanism of the spin diffusion anisotropy in ferro- and ferrimagnets, which originates mainly from the long-range dipole-

dipole interaction[59]. In antiferromagnets and in weak ferromagnets like hematite and orthoferrites, the net magnetization is small and the dipole-dipole interactions can thus be neglected.

We further discuss the role of magnetoelastic coupling or magnetostriction on anisotropy in spin transport. We would like to point out that indeed orthoferrites are known for the pronounced magnetoelastic coupling that is responsible, for example, for the emission of the acoustic waves by a moving domain wall[61,62] or for the anisotropy of the domain wall motion.

It is expected that the coupling between magnon and acoustic phonon modes mediated by magnetoelastic interactions can contribute to the anisotropy of the magnon transport and large propagation length. To check the relevance study of this issue, we analyzed coupled magnon-phonon dynamics in the presence of an external magnetic field. We consider an experimentally relevant geometry with excitations propagating either along the $a$ or $c$ axis.

First, we note that the limiting magnon velocity (~20-25 km/s) is larger than the speed of the transverse ($s_t \approx 4$ km/s) and longitudinal ($s_l \approx 7$ km/s) acoustic phonons[61,62]. This excludes crossing points of magnon and phonon spectra and predicts stronger hybridization with the longitudinal phonon mode (compared to the transversal mode). However, the coupling with the longitudinal phonons, $\lambda n_x^{(0)} n_y^{(0)}$, (where $\lambda$ is a magnetoelastic constant, $\lambda = \lambda_{11} - \lambda_{12}$ for $\boldsymbol{k} \| a$ and $\lambda = \lambda_{21} - \lambda_{22}$ for $\boldsymbol{k} \| c$) depends on the orientation of the Néel vector and attains its maximal values at $\mu_0 H \propto 3$T. Further, hybridization between the magnons and phonons reaches a maximum for wave-vectors $k \approx k_0$, for which the group velocity of magnons, and the velocity of longitudinal phonons coincide, $v = d\omega_1(k_0; H)/dk \approx s_l$. Such hybridization allows for spin transport with phonons and gives potentially rise to a larger relaxation time $\tau$.

Second, according to ab initio calculations[63], the elastic modulus of YFeO$_3$ shows a strong anisotropy. We estimate the difference between $s_l(\boldsymbol{k} \| a)$ and $s_l(\boldsymbol{k} \| c)$ to be 20%. This also means a 20% difference between the velocity of hybridized magnons with $k \approx k_0$ propagating along the $a$ and $c$ axis. Assuming that the spin signal is transported mainly by these particular magnons, we anticipate an additional splitting of the magnon group velocities $v_x$ and $v_y$. However, while magnetoelastic coupling can play a role, we do not probe the impact of varying strain on the spin transport. On the other hand, this is of course of interest for a future study.

Regarding the effect of ferroelectricity on spin transport, we would like to point out that reports for ferroelectricity and piezoelectricity exist in hexagonal phase of YFeO$_3$[19,21]. The electronic origin of ferroelectricity is attributed to the asymmetric $Y 4d_z^2 - O_A 2p_z$ hybridization. On the other hand, in the bulk orthorhombic centro-symmetric phase of YFeO$_3$, theoretically, ferroelectricity is in fact prohibited. But recently very weak ferroelectric polarization more than three order of magnitude lower has also reported in bulk orthorhombic phase[36]. The origin of strong ferroelectricity was revealed in thin films of YFeO$_3$ in its orthorhombic centrosymmetric phase due to a Y-Fe anti-site defect mechanism[20]. In this study, we use a YFeO$_3$ bulk single crystal sample and the x-ray diffraction measurements establish the orthorhombic phase. Hence, we expect as a result no significant (or at least very weak) ferroelectric polarization in this sample and thus no significant effect of it on spin transport. However, it is an interesting property that can be explored in the future going beyond the scope of this current work.

The presence of interfacial DMI in Y$_3$Fe$_5$O$_{12}$ (YIG, a ferrimagnetic insulator), generated at YIG/Gd$_3$Ga$_5$O$_{12}$ (GGG) interface, induces a chiral group velocity[64] and magnon drift current[65]. Such interfacial DMI originates from the inversion symmetry breaking in the film normal direction and spin-orbit coupling. But in YFeO$_3$ (in single crystal form) the DMI originates from the exchange interaction between the two sublattices, which generate a small canting between two sublattices and consequently, a net magnetic moment is induced perpendicular to Néel vector. Such DMI has no effect on spin-wave group velocity. The

DMI that is responsible for the chiral group velocity in YIG is related with so-called Lifshitz invariants and this is forbidden in YFeO$_3$ due to the space inversion symmetry.

The observed long-distance spin transport in the ultra-low damping, insulating canted antiferromagnet YFeO$_3$ is an important step forward towards the ultimate goal of establishing a universal model for long distance spin transport in antiferromagnetic insulators. Our findings open up a large and important class of low damping materials in which one can transport spin information. The transport mechanism previously identified in hematite is not universal to all antiferromagnets and entail limitations: in easy-axis antiferromagnets, only intrinsic circularly polarized eigenmodes can carry spin angular momentum and in easy-plane antiferromagnets, a superposition of two linearly polarized eigenmodes can support spin transport, but the transport length scales are dominated by the dephasing length. The reported mechanism in YFeO$_3$ allows us to transport spin by modifying the ellipticity of magnon modes. Hence, we emphasize the fact that all the antiferromagnets with a low magnetic damping have the potential to transport spin if one tunes the ellipticity of the magnon eigenmodes. For this, there are different possible approaches and here we demonstrate that we can use a field to modify the ellipticity in the presence of strong DMI. Also, potentially, strain and ferroelectricity (if present) could additionally be used to tailor the magnon properties. Finally, we point out that whether the presence of strong DMI is essential to efficiently modify the magnon ellipticity with an applied field is an open question that warrants study in further materials. We calculate the value of $\lambda$ at 5.5 T and 3 T (shown in Fig. 3), as in such field R$_{el}$ shows a maximum in the device geometries shown in Fig. 2a, b, respectively. It was demonstrated in YIG that $\lambda$ increases with decreasing the magnetic field[55]. In the same analogy, in YFeO$_3$, we also expect $\lambda$ to increases with decreasing magnetic field further and an efficient spin transport can be realized at very low field and at room temperature (as a significant spin transport signal exists in 300 K, shown in Supplementary Fig. 14). Furthermore, the observed mode of transport demonstrated by nonlocal transport measurements over long distances along with the antiferromagnetic resonance measurements highlights the potential of low damping antiferromagnetic insulators for their integration into next generation magnonic and spintronic devices.

## Methods
### Single crystal preparation
The YFeO$_3$[010] crystal is prepared by optical floating zone technique from the sintered polycrystalline sample. The crystallographic properties have been investigated by x-ray diffraction, which confirm the crystallinity of the crystal and also the angle of miscut between the crystallographic axes and sample plane (see Supplementary Fig. 3). The details of the growth and characterization are discussed in the supplementary. The magnetization hysteresis curve is measured using superconducting quantum interference device to detect the weak magnetic moment along the intermediate anisotropy $c$-axis.

### Non-local transport measurements
Prior to the pattering, the sample was cleaned with acetone, isopropanol and deionized water to remove any organic surface residue. The devices were patterned using e-beam lithography followed by sputter deposition of 7-nm Pt and lift-off. The electrical contact pads were defined by a second e-beam lithography step followed by the deposition of Cr (5 nm)/Au (45 nm) and lift-off. Devices are consisting of three wires of length 50 μm and width 300 nm. The center-to-center separation between the wires varies from 500 nm to 2.5 μm. A scanning electron microscope (SEM) image of a typical device is shown in Fig. 2c. The sample was mounted to a piezo-rotating element in a variable temperature insert installed in a superconducting magnet, which is capable of fields up to 12 Tesla (T). We pass a charge current

through the central wire with a current density of $3.8 \times 10^7 \, A/cm^2$ and measure a nonlocal voltage at the detector wires. The nonlocal voltage is recorded for positive and negative polarity of current as a function of field, spatial distance between the wires and the angle between current and field directions.

## Data availability

The data that support the findings of this study are available from the corresponding authors upon reasonable request. Correspondence and material request should be addressed to M.K. or S.X.C.

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

## Acknowledgements

S.D. thanks Mr. T. Reimer of Johannes Gutenberg University Mainz for his help in fabricating the devices. This work was supported by the Max Planck Graduate Center with the Johannes Gutenberg-Universität Mainz (MPGC). The authors in Mainz acknowledge support from the DFG project number 423441604. R.L. and M.K. acknowledge financial support from the Horizon 2020 Framework Programme of the European Commission under FET-Open grant agreement no. 863155 (s-Nebula). All authors from Mainz also acknowledge support from both MaHoJeRo (DAAD Spintronics network, project number 57334897 and 57524834), SPIN + X (DFG SFB TRR 173 No. 268565370, projects A01, A03, A11, B02, and B12), and KAUST (OSR-2019-CRG8-4048.2). M.K. acknowledges support by the Research Council of Norway through its Centers of Excellence funding scheme, project number 262633 "QuSpin" and the Horizon Europe Framework Programme of the European Commission under grant agreement no. 1010702P7 (SWAN-om-chip). O.G. and J.S. additionally acknowledge support from the ERC Synergy Grant SC2 (No. 610115), EU FET Open RIA Grant no. 766566, and funding from Deutsche Forschungsgemeinschaft (DFG) via" TRR 288 – 422213477 (Projects No. A09). J.S. additionally acknowledges TopDyn JGU Grant. S.B acknowledges the Deutsche Forschungsgemeinschaft (DFG, German Research Foundation)—project number 358671374. S.X.C. acknowledges support by the Science and Technology Commission of Shanghai Municipality (No.21JC1402600), and the National Natural Science Foundation of China (NSFC, No. 12074242). R.L. acknowledges financial support from the Horizon 2020 Framework Programme of the European Commission under FET-Open grant agreement No. 964931 (TSAR).

## Author contributions

M.K. and A.R. conceived the idea. X.X.M. and H.Y.C. grew, cut, and characterized the YFeO₃ single crystals used in the experiments under the guidance of S.X.C. S.B. and S.D. performed the structural and magnetic characterization. S.D. fabricated the devices and M.A.S. deposited Pt. The non-local transport measurements are performed by S.D. with the help of S.B., C.S. E.F.G. and F.F. Magnetic resonance measurements are performed by F.V.D., U.E., V.B., A.L.B. and R.L. The theoretical model is developed by O.G. The data are analyzed by S.D., M.K., A.R., R.L., S.B., O.G., J.S. and G. J. The manuscript is written by S.D. with input from A.R. O.G. R.L and M.K. All the authors revised the manuscript.

## Funding

## Competing interests

The authors declare no competing interests.
