## [Peer Review File · Nature Communications]

Anisotropic long-range spin transport in canted antiferromagnetic orthoferrite YFeO₃Reviewer #1 (Remarks to the Author):

The manuscript "Anisotropic long-range spin transport in canted antiferromagnetic orthoferrite YFeO₃" by Shubhankar Das et al. presents long-distance incoherent magnon transport in canted antiferromagnetic orthoferrite YFeO₃. By using the electron paramagnetic resonance spectrometer, ultra-low damping of about 9×10^{-5} is obtained based on a linear fitting. Two series of devices whose Pt wires are patterned along both in-plane easy and hard axes demonstrated the anisotropic magnon decay length, which mainly attribute to the the anisotropic magnon group velocities along easy and hard axes. The different magnon transport feature in this materials from previous well-studied Fe₂O₃ is considered contributing from the three factors magnon density, frequency, and ellipticity both depending on magnetic field. The observed features are interesting and could have an impact on incoherent magnonics. The experimental method and device design used in this work are very similar to those in their previous works such as Ref. 5. However, the experiments are carried out very systematically with different orientations and transport distances. The material of YFeO₃ exhibits a remarkably low damping making it the „YIG“ of antiferromagnets and very much worth investigating. It is also appreciated that the authors have studied theoretically to account for the detected DC transport signals, where the theoretical fitting agrees excellently well with the experimental data. The explanation of anisotropic decay length related on different exchange stiffness and group velocities is reasonable and applaudable.

This manuscript may be suitable for publication in Nature Communications. However, I have some major comments which should be addressed for a clear presentation:

1. First of all, the authors sometimes describe the observed phenomena as „diffusive magnon transport“ and sometimes as „propagation of magnons“. In fact the word „propagation“ imply a coherent transport/propagation of waves, where the wavelengths and frequencies of magnons should be well defined. However, the work presented by the authors clearly investigate the incoherent transport of magnons with a diffusive nature as the authors stated in line 102 on Page 5 and also line 151 on Page 6. The magnons that are transported follow the Bose distribution, which indicate that the wavevector k is not well defined, but more in a statistic regime. Therefore, I would strongly recommend that the authors unify the terminology in describing the transport of magnons which should reveal its diffusive nature rather than a coherent propagation nature (which may be misleading for the readers).

2. For the damping characterization, I find that the error bars and differences of the last two points with higher frequency are quite large when comparing with other points in Fig. 1(b). In addition, in the supplementary material, the excitation frequency is fixed at 452 GHz and the resonance field is about 7.2 T. However, in the Fig. 4(a) in the main text, there is not a corresponding mode showing in the field-dependent frequencies. I suppose that more data points and detailed explanation would be helpful for optimizing the obtained value and it would be better to show all the raw data at each frequencies in the supplementary.

The canting in YFeO₃ is attributed to the large DMI value. Recently, the DMI has been also observed in the ferromagnetic YIG, which generates chiral group velocities or drift group velocities, such as:

[1]. Richard Schlitz et. al. Control of Nonlocal Magnon Spin Transport via Magnon Drift Currents. Phys. Rev. Lett. 126, 257201 (2021).

[2]. H. Wang et. al. Chiral Spin-Wave Velocities Induced by All-Garnet Interfacial Dzyaloshinskii-Moriya Interaction in Ultrathin Yttrium Iron Garnet Films. Phys. Rev. Lett. 124, 027203 (2020).

Therefore, I would suggest that the authors discuss the influence of DMI on spin-wave group velocities which are essential to explain their results.

3. Compared with hematite, the YIG shows very low damping, but unfortunately can only transport spin information at very high magnetic field 5 T or 3 T in Fig. 2 and also at low temperature of 200 K. Therefore, it is very hard for its practical applications as claimed in the introduction and summary. The authors should comment on this and provide some outlooks on how to go beyond this.

4. The authors should mention and discuss some earlier work on the characterization of

YFeO₃, since it is not a new material. For example, Yamaguchi et al. Phys. Rev. Lett. 105, 237201 (2010). For magnon decay lengths, the authors should mention some in other material systems, such as ferromagnetic insulators (Maendl, S. et al. Appl. Phys. Lett. 111, 012403 (2017).), metallic oxides (Qin, Q. et al. Appl. Phys. Lett. 110, 112401 (2017). and Liu, C. et al. Nat. Nanotechnol. 14, 691–697 (2019).) and also very recently in AFM/FM multiferroic heterostructures (J. Zhang et al. Nat. Commun. 12, 7258 (2021)).

And also some minor comments/suggestions as the following,

1) The damping parameter is estimated from the linear fit in Fig. 1b, which yields „9.3 $\mu\text{m}^2 \times 10^{-5}$ “. If the error bar is already 2.0, the 0.3 may be inaccurate and not necessary. Same for 525 μm 50 nm in Line 223 Page 9.

2) The legend and inset in Fig. 3 is too small to read. The authors should either enlarge the front size to the same of „Distance (nm)“ or try describe them in captions.

3) It would be better to add detailed SFig. number in the main text when the authors point the information to the supplementary for readers' convenience.

4) The unit Tesla should either go with „B“ or „ $\mu_0 H$ “

5) Some language polishing is needed, e.g. „over large distances“ at Line 100 should better be „over long distances“. „more apt“ at line 121 may be better „apter“ or „more suitable for device applications“

Reviewer #2 (Remarks to the Author):

The manuscript discusses long-distance transport of antiferromagnetic magnons in orthoferrite YFeO₃. The authors show anisotropic magnon decay lengths, attributed to the change in the magnon group velocity that arises from different exchange stiffness. In the manuscript, the authors compare magnon properties in YFeO₃ to those in hematite, a low-damping antiferromagnetic material. They find interesting magnon properties in YFeO₃ including non-zero field transport, nearly constant antiferromagnetic anisotropy over a broad temperature range, and a unique spin transport mode, which make antiferromagnetic YFeO₃ promising for magnonic devices. Overall, these results are interesting and original, but I don't find enough novelty of the manuscript.

In my views, zero external field is good for the operation of a magnonic device since it is more energy-efficient and CMOS-compatible. In this sense, hematite is better than YFeO₃. Regarding the damping of YFeO₃, I find the authors address the damping constant of 9.3×10^{-5} based on the poor fit of the data to Fink model. Also, the inhomogeneous broadening seems rather large, which leads to large effective damping and short decay lengths. For device fabrication, the authors patterned the Pt wires on top of the bulk crystal, not on the thin film that is relevant for technological applications. The authors mention a bit the effect of DMI on the spin transport, which is interesting, but I don't see much experimental evidence in the manuscript. The authors mention YFeO₃ exhibits interesting properties such as piezo-electricity and large magnetostriction, fast domain wall motion, which might be useful for the control of antiferromagnetic magnons, again I don't see any experimental data or discussion. All in all, the manuscript reports an incremental advance in antiferromagnetic magnonics rather than breakthroughs in physics, devices, and technologies of magnonics. Therefore, I cannot recommend for publication in Nature Communications.

Our response to Reviewer # 1

Comment: The manuscript “Anisotropic long-range spin transport in canted antiferromagnetic orthoferrite YFeO₃” by Shubhankar Das et al. presents long-distance incoherent magnon transport in canted antiferromagnetic orthoferrite YFeO₃. By using the electron paramagnetic resonance spectrometer, ultra-low damping of about 9×10^{-5} is obtained based on a linear fitting. Two series of devices whose Pt wires are patterned along both in-plane easy and hard axes demonstrated the anisotropic magnon decay length, which mainly attribute to the anisotropic magnon group velocities along easy and hard axes. The different magnon transport feature in this material from previous well-studied Fe₂O₃ is considered contributing from the three factors magnon density, frequency, and ellipticity both depending on magnetic field. The observed features are interesting and could have an impact on incoherent magnonics. The experimental method and device design used in this work are very similar to those in their previous works such as Ref. 5. However, the experiments are carried out very systematically with different orientations and transport distances. The material of YFeO₃ exhibits a remarkably low damping making it the “YIG” of antiferromagnets and very much worth investigating. It is also appreciated that the authors have studied theoretically to account for the detected DC transport signals, where the theoretical fitting agrees excellently well with the experimental data. The explanation of anisotropic decay length related on different exchange stiffness and group velocities is reasonable and applaudable. This manuscript may be suitable for publication in Nature Communications. However, I have some major comments which should be addressed for a clear presentation.

Our response: We thank the reviewer for the thorough assessment of our manuscript and the support for publication in Nature Communications. We thank the reviewer in particular for the detailed comments that have allowed us to improve our manuscript. Our responses to the reviewer’s comments are detailed below.

Major comments:

Comment 1: First of all, the authors sometimes describe the observed phenomena as “diffusive magnon transport” and sometimes as “propagation of magnons”. In fact the word “propagation” imply a coherent transport/propagation of waves, where the wavelengths and frequencies of magnons should be well defined. However, the work presented by the authors clearly investigate the incoherent transport of magnons with a diffusive nature as the authors stated in line 102 on Page 5 and also line 151 on Page 6. The magnons that are transported follow the Bose distribution, which indicate that the wavevector k is not well defined, but more in a statistic regime. Therefore, I would strongly recommend that the authors unify the terminology in describing the transport of magnons which should reveal its diffusive nature rather than a coherent propagation nature (which may be misleading for the readers).

Our response: We thank the reviewer for this valid comment that we completely agree with. The transport of magnons in the studied material (YFeO₃) is here incoherent in nature and the transport is diffusive. Hence, we change in the revised manuscript the terminology to clearly convey our findings.

#Comment 2: For the damping characterization, I find that the error bars and differences of the last two points with higher frequency are quite large when comparing with other points in Fig. 1(b). In addition, in the supplementary material, the excitation frequency is fixed at 452 GHz and the resonance field is about 7.2 T. However, in the Fig. 4(a) in the main text, there is not a corresponding mode showing in the field-dependent frequencies. I suppose that more data points and detailed explanation would be helpful for optimizing the obtained value and it would be better to show all the raw data at each frequency in the supplementary.

Our response: The method used to determine damping and error bars has been refined and clarified in the revised manuscript.

We use a wavelength in the same order as the sample and because of that, at low fields there is an additional linewidth broadening due to the presence of magneto-static modes. In our previous method, we took the linewidth of all visible peaks, which does not take the linewidth broadening into account and that several peaks can overlap. From those measurements, we could deduce a magnetic damping coefficient with an upper bound of 9×10^{-5} .

For our refined method as detailed in the revised manuscript, we extract the average linewidth from the smallest peak-to-peak distances of the resonances, which relate to the damping coefficient. These distances are indicated in the new supplementary figures Fig. S7 to S10. The error bars are determined by the maximum deviation of the average value.

Resonance data have now been measured for the two cases discussed in the manuscript: $H \parallel a$ (easy axis) and $H \parallel c$ (intermediate axis). The linewidth dependence is fitted using the theoretical model of antiferromagnetic resonance provided by Fink (Ref. 41 of the manuscript, please see revised Fig. 1(b) of main text and Fig. S6 of the supplementary), which yields α_G of $3.5 \pm 0.4 \times 10^{-6}$ and $6.2 \pm 0.3 \times 10^{-6}$ at 20 K and 150 K, respectively, for field along c-axis and $6 \pm 2 \times 10^{-6}$ at 20 K for field along a-axis. We would like to point out that this still corresponds to an upper bound for the magnetic damping.

The corresponding frequency dependent resonance field is now shown in Fig. S5, along with a fitting curve by use of the model described in the revised manuscript.

All the raw data have been added to the supplemental material (see new Figures S7 to S10).

Note: in the graphs, the linewidth now accounts for the FWHM to avoid misleading the reader. Previous data accounting for the peak-to-peak data have been multiplied by $\sqrt{3}$.

Also, previous measurements were done at 150K, compared to 20K for current measurements. The frequency dependent resonance field follows the same fit, however the damping is approximately a factor two lower for 20K compared to 150K.

We have recalculated the field dependence of eigen-frequencies (Fig. 4(a)) by using the experimental parameters used to fit the AFMR data shown in Fig. S5 of supplementary.

#Comment 3: The canting in YFeO₃ is attributed to the large DMI value. Recently, the DMI has been also observed in the ferromagnetic YIG, which generates chiral group velocities or drift group velocities, such as:
[1]. Richard Schlitz et. al. Control of Nonlocal Magnon Spin Transport via Magnon Drift Currents. Phys. Rev. Lett. 126, 257201 (2021).
[2]. H. Wang et. al. Chiral Spin-Wave Velocities Induced by All-Garnet Interfacial Dzyaloshinskii-Moriya Interaction in Ultrathin Yttrium Iron Garnet Films. Phys. Rev. Lett. 124, 027203 (2020).
Therefore, I would suggest that the authors discuss the influence of DMI on spin-wave group velocities which are essential to explain their results.

Our response: The referee is of course correct that the chiral group velocity and magnon drift current in YIG films likely originates from interfacial DMI generated at the YIG/GGG interface. Such interfacial DMI stems from the inversion symmetry breaking in the film normal direction and spin-orbit coupling. But in YFeO₃ (in single crystal form) the DMI originates from the exchange interaction between the two sublattices, which generate a small canting between two sublattices and consequently, a net magnetic moment is induced perpendicular to Néel vector. Such DMI has no effect on spin-wave group velocity. The DMI that is responsible for the chiral group velocity in YIG is related with “Lifshitz invariants” and this is forbidden in YFeO₃ due to the space inversion symmetry.

To make this clear, we have incorporated the above discussion in the revised manuscript and we cite the references suggested by the referee in our revised manuscript.

#Comment 4: Compared with hematite, the YIG shows very low damping, but unfortunately can only transport spin information at very high magnetic field 5 T or 3 T in Fig. 2 and also at low temperature of 200 K. Therefore, it is very hard for its practical applications as claimed in the introduction and summary. The authors should comment on this and provide some outlooks on how to go beyond this.

Our response: Fundamentally the magnon eigenmodes in YFeO₃ at zero applied magnetic field are linearly polarized and, consequently, cannot carry spin angular momentum. But, by the application of an external magnetic field along the easy axis, the Néel vector rotates under the presence of Dzyaloshinskii-Moriya interaction (DMI) field, and the elliptically polarized magnon eigenmodes are stabilized only when the Néel vector has a finite projection on the magnetic field direction. Such elliptically polarized magnons carry the spin information efficiently. We calculate the magnon decay length (λ) at 5.5 T and 3 T (shown in Fig. 3 of the revised manuscript), as in such field the electrical spin transport signal (R_{el}) shows a maximum in the device geometries shown in Fig. 2(a) and (b), respectively. It was demonstrated in YIG (a ferrimagnetic insulator) that the λ increases with decreasing the magnetic field (L. J. Cornelissen et al., Phys. Rev. B 93, 020403(R) (2016)). We also expect λ to increase with decreasing magnetic field further and efficient spin transport can be realized at very low field. Although zero field transport is prohibited in YFeO₃ due to orthorhombic symmetry, but the new mode of transport opens up a huge and technologically relevant class of low damping orthoferrites with various symmetries to spin transport for further studies. Finally, one should

note that of course applying a magnetic field permanently by adding locally permanent magnets (or just hard magnetic microstructures) on the chip is an option that might make this mode of transport accessible also on the device level.

While we have performed the transport measurements at 200 K, efficient spin transport signals exist up to room temperature. Fig. S15 of supplementary shows nonlocal electrical spin transport signal as a function of temperature between 50 K to 300 K.

So given these findings we can envisage that our identified transport mode can be used for applications and this is discussed in the revised manuscript.

Comment 5: The authors should mention and discuss some earlier work on the characterization of YFeO₃, since it is not a new material. For example, Yamaguchi et al. Phys. Rev. Lett. 105, 237201 (2010). For magnon decay lengths, the authors should mention some in other material systems, such as ferromagnetic insulators (Maendl, S. et al. Appl. Phys. Lett. 111, 012403 (2017).), metallic oxides (Qin, Q. et al. Appl. Phys. Lett. 110, 112401 (2017). and Liu, C. et al. Nat. Nanotechnol. 14, 691–697 (2019).) and also very recently in AFM/FM multiferroic heterostructures (J. Zhang et al. Nat. Commun. 12, 7258 (2021)).

Our response: We thank the referee for highlighting these references and we have incorporated these in the discussions of some previous works in the revised manuscript.

Minor Comments:

Comment 6: The damping parameter is estimated from the linear fit in Fig. 1b, which yields “ $9.3 \pm 2 \times 10^{-5}$ ”. If the error bar is already 2.0, the 0.3 may be inaccurate and not necessary. Same for 525 ± 50 nm in Line 223 Page 9.

Our response: We have corrected the fitting of antiferromagnetic resonance linewidth vs frequency curve (Fig. 1(b)) and these have been incorporated in the revised manuscript.

Comment 7: The legend and inset in Fig. 3 is too small to read. The authors should either enlarge the front size to the same of “Distance (nm)” or try describe them in captions.

Our response: We have now mentioned the value of λ in the figure caption.

Comment 8: It would be better to add detailed SFig. number in the main text when the authors point the information to the supplementary for readers’ convenience.

Our response: We have done the change according to this helpful suggestion.

Comment 9: The unit Tesla should either go with “B” or “ μ_0 H”.

Our response: We have changed the field sign.

Comment 10: Some language polishing is needed, e.g. “over large distances” at Line 100 should better be “over long distances”. “more apt” at line 121 may be better “apter” or “more suitable for device applications”.

Our response: We have gone through the manuscript in detailed and done appropriate modifications with the support of a native speaker.

Our response to Reviewer # 2

Comment 1: The manuscript discusses long-distance transport of antiferromagnetic magnons in orthoferrite YFeO_3 . The authors show anisotropic magnon decay lengths, attributed to the change in the magnon group velocity that arises from different exchange stiffness. In the manuscript, the authors compare magnon properties in YFeO_3 to those in hematite, a low-damping antiferromagnetic material. They find interesting magnon properties in YFeO_3 including non-zero field transport, nearly constant antiferromagnetic anisotropy over a broad temperature range, and a unique spin transport mode, which make antiferromagnetic YFeO_3 promising for magnonic devices. Overall, these results are interesting and original, but I don't find enough novelty of the manuscript.

Our response: We thank the reviewer for giving his valuable suggestions to improve the manuscript. While Reviewer 1 supports publication of a revised manuscript in Nature Communications, Reviewer 2 has raised concerns about the novelty of our results. We strongly refute the notion that there is not enough novelty in our work. While the referee mentions some of our results, we are not sure that we have sufficiently well highlighted the novel and important findings. So, in response, we have listed the important highlights of the work and explain the novel direction on spin transport based on antiferromagnetic materials that are enabled by our study, including important findings not mentioned by the reviewer.

- Although the magnetic damping is similarly low to hematite, the transport mechanisms is qualitatively different from that reported for hematite. A key difference is that unlike in hematite, where both the Néel vector and the weak canted moment can contribute, we find here that the transport mediated by the Néel vector dominates in all cases and this results from the transport mechanism detailed next.
- Different from hematite, the magnon eigenmodes are non-degenerate at zero field and spin angular momentum is transported by an elliptically polarized magnon eigenmode, which is stabilized by the presence of Dzyaloshinskii-Moriya interaction (DMI) and an external magnetic field. This leads to qualitatively novel transport regimes that are reflected for instance in unique field dependences that are distinct from established hematite behaviour.
- We find that the magnon decay length depends strongly on the relative orientation of the applied field and transport directions. The anisotropic magnon decay length highlights the importance of the magnon group velocity in the propagation of spin waves in this class of materials. The dependence of the magnon decay length on magnon group velocity provides an additional handle to tune the magnon transport properties in non-local devices.
- Our results are an important step forward in establishing a universal understanding and a model that can explain long-distance spin transport in antiferromagnetic insulators of various symmetries.
- Orthoferrites belong to the wide class of materials whose properties can be tuned by the proper choice of RE element. The demonstrated novel mode of transport in YFeO_3 , opens up the huge and technologically relevant class of low-damping, canted antiferromagnetic orthoferrites to long-distance spin transport.

- Orthoferrites are the class of materials having mainly orthorhombic structure and the orthorhombicity excludes antiferromagnetic domains as we only have only one easy axis. This leads to less scattering of magnonic spin current from domain walls, yielding a longer magnon decay length so that fundamentally this class can outperform other established materials.

Comment 2: In my views, zero external field is good for the operation of a magnonic device since it is more energy-efficient and CMOS-compatible. In this sense, hematite is better than YFeO₃.

Our response: We completely agree with the reviewer that zero field transport is always desirable for applications. However, it is relatively straight forward (and realized in many industrial sensor devices that are pre-biased) to implement a constant magnetic field generated by a permanent magnet structure (or hard magnetic microstructures) realized on chip that permanently generates the optimal field for the transport.

Furthermore, we want to emphasize the fact that the transport mode in hematite is not universal and not applicable to most of the antiferromagnetic materials. Whereas in case of YFeO₃, the elliptically polarized magnons, which carry spin angular momentum, can be stabilized by external magnetic field in presence of DMI. Such unique transport mode may be applicable to wider range of antiferromagnetic materials with various symmetries. Other advantages of transport mode of YFeO₃ have been listed in our response to comment 1.

Comment 3: Regarding the damping of YFeO₃, I find the authors address the damping constant of 9.3×10^{-5} based on the poor fit of the data to Fink model. Also, the inhomogeneous broadening seems rather large, which leads to large effective damping and short decay lengths.

Our response: We thank the referee for raising this valid point. In response we have done extensive additional analysis and we have in particular refined the method used to determine the damping and error bars have been added as clarified in the revised manuscript. Reviewer 1 has also commented about this, so see also our response to the #Comment 2 of Reviewer 1 for additional details.

In YFeO₃, for H parallel to c-axis, the lowest value for the frequency, i.e., the value of the gap is related to the material's intrinsic parameters: H_{ex} and H_a, and not to inhomogeneous broadening. It is given by $f = \frac{\gamma}{2\pi} \sqrt{H_{ex}H_a} \sim 300 \text{ GHz}$.

Comment 4: For device fabrication, the authors patterned the Pt wires on top of the bulk crystal, not on the thin film that is relevant for technological applications.

Our response: We agree with the reviewer that for technological application realisation of thin films is important. We started with single crystal intentionally as in case of hematite single crystal samples are used to demonstrate the transport mechanisms (Lebrun et al., Nature **561**, 222-225 (2018)) before thin film samples are then employed to demonstrate applicability (Ross

et al., Nano Lett. **20**, 306-313 (2019)). Furthermore, due to the larger magnetic domain size in single crystal samples, these had been found to show larger spin transport length scales. The reduced magnon decay length in thin film samples attributed to the scattering of magnonic spin current with domain walls. We are aware of the fact that the group of Prof. Caroline Ross (Ning et al., Nat. Commun. **12**, 4298 (2021)) has successfully deposited YFeO₃ epitaxial thin film and it shows room temperature ferroelectricity due to antisite defect mechanism. Given the demonstrated high quality of the thin films, for the future we plan to perform spin transport experiments in YFeO₃ thin films and also to investigate the influence of ferroelectric transition on magnon spin transport. However, such work is clearly beyond the scope of the presented manuscript.

Comment 5: The authors mention a bit the effect of DMI on the spin transport, which is interesting, but I don't see much experimental evidence in the manuscript.

Our response: In YFeO₃, the DMI originates from the exchange interaction between the two sublattices, which generate a small canting between two sublattices and consequently, a net magnetic moment is induced perpendicular to Néel vector. The combined effect of DMI and external magnetic field stabilizes the magnons in elliptically polarized mode. But such DMI has no effect on spin wave group velocity and consequently on magnon spin transport. Please see our response to the #comment 3 of Reviewer 1 for a detailed discussion.

Comment 6: The authors mention YFeO₃ exhibits interesting properties such as piezo-electricity and large magnetostriction, fast domain wall motion, which might be useful for the control of antiferromagnetic magnons, again I don't see any experimental data or discussion.

Our response: In our revised manuscript we now discuss these interesting properties that are observed in orthoferrites with different rare-earth elements in more detail. And it will be exciting to investigate the effect of piezo-electricity and magnetostriction on spin transport in this class of materials. Hence, this highlights the importance of our study on this exciting material further. However, investigating the effect of these individual mechanisms on spin transport is separate study by itself and beyond the scope of current manuscript.

Reviewer #1 (Remarks to the Author):

The authors have addressed my comments and questions adequately with detailed explanation and additional efforts. In general, I find that this experimental work was carried out systematically, well written and contribute to the emerging field of antiferromagnetic magnonics. Thus from my side, I can now recommend its publication in Nature Communications.

Reviewer #2 (Remarks to the Author):

In the response letter and revised manuscript, the authors list their important findings and emphasize the novelty of the manuscript. I appreciate the authors' efforts. But unfortunately, I don't find significant improvement in the revised manuscript. The results presented in the manuscript are novel and original, but are not enough for the publication in Nature Communications, as I mentioned before. I suggested the authors include some new experimental results or discussions on the effects of piezoelectricity or large magnetostriction in YFeO₃ on magnon properties. But the authors don't take my point.

Again, I concern the accuracy of the damping constant of YFeO₃ extracted from the experimental data. The authors present the damping constant of $(3 - 6) \times 10^{-6}$, which seems the lowest value reported so far, to my knowledge. After checking the raw data, I am not sure how reliable the value is. I suggest the authors should carefully analyze the data.

Our response to Reviewer #2

Comment 1: In the response letter and revised manuscript, the authors list their important findings and emphasize the novelty of the manuscript. I appreciate the authors' efforts. But unfortunately, I don't find significant improvement in the revised manuscript. The results presented in the manuscript are novel and original, but are not enough for the publication in Nature Communications, as I mentioned before. I suggested the authors include some new experimental results or discussions on the effects of piezoelectricity or large magnetostriction in YFeO_3 on magnon properties. But the authors don't take my point.

Our response: We thank the reviewer for acknowledging the novelty and originality of our results. While Reviewer #1 recommends publication of our revised manuscript, this reviewer has asked for additional information. We appreciate reviewer's comments and suggestions for additional discussion on the effects of piezoelectricity and large magnetostriction in YFeO_3 on magnon properties. We gladly take up this suggestion and our responses to these points are below.

Firstly, concerning the piezoelectricity, we would like to point out that reports for ferroelectricity and piezoelectricity exist in **hexagonal** phase of YFeO_3 . In a number of papers, such as for instance in S.-J. Ahn et al., *Materials Chemistry and Physics* 138, 929e936 (2013) and R. Zhang et al., *Journal of Electroceramics* 40, 156-161 (2018), these properties were studied. The electronic origin of ferroelectricity is attributed to the asymmetric $Y4d_z^2 - O_A 2p_z$ hybridization. On the other hand, in the bulk **orthorhombic** centro-symmetric phase of YFeO_3 , theoretically, ferroelectricity is in fact prohibited. But recently very weak ferroelectric polarization more than three order of magnitude lower is also reported in the orthorhombic phase (M. Shang, et al., *Appl. Phys. Lett.* 102, 062903 (2013)). Recently, the origin of strong ferroelectricity was revealed in thin films of YFeO_3 in its orthorhombic centrosymmetric phase due to a Y-Fe anti-site defect mechanism [S. Ning et al., *Nature Commun.* 12, 4298 (2021)]. In our case, we use a YFeO_3 bulk single crystal sample and the x-ray diffraction measurements establish the orthorhombic phase. Hence, we expect as a result no significant (or at least very weak) ferroelectric polarization in our sample and thus no significant effect of it on spin transport.

However, we now discuss this in more detail in the revised manuscript as it is an interesting property that can be explored in the future going beyond the scope of our current work, which focuses on spin transport where we do not study any impact of possible ferro- or piezoelectricity.

Secondly, concerning the magnetostriction, we would like to point out that indeed orthoferrites are known for the pronounced magnetoelastic coupling that is responsible, for example, for emission of the acoustic waves by a moving domain wall [V. G. Bar'yakhtar, et al., *Soviet Physics Uspekhi*, 28, 563–588 (1985) and J. Krzywiński, *Journal of Magnetism and Magnetic Materials* 59, 62–68 (1986).] or for an anisotropy of the domain wall motion.

We believe that the coupling between magnon and acoustic phonon modes mediated by magnetoelastic interactions can contribute to the anisotropy of magnon transport and large propagation length. To check the relevance of this issue, we analyzed coupled magnon-phonon

dynamics in presence of the external magnetic field. We consider the experimentally relevant geometry with excitations propagating either along the a or c axis.

First, we note that the limiting magnon velocity ($\sim 20\text{-}25$ km/s) is larger than the speed of the transverse ($s_t \approx 4$ km/s) and longitudinal ($s_l \approx 7$ km/s) acoustic phonons [V. G. Bar'yakhtar, et al., *Soviet Physics Uspekhi*, 28, 563–588 (1985) and J. Krzywiński, *Journal of Magnetism and Magnetic Materials* 59, 62–68 (1986)]. This excludes crossing points of the magnon and phonon spectra and predicts stronger hybridization with the longitudinal phonon mode (compared to the transversal mode). However, the coupling with the longitudinal phonons, $\lambda n_x^{(0)} n_y^{(0)}$, (where λ is a magnetoelastic constant, $\lambda = \lambda_{11} - \lambda_{12}$ for $\mathbf{k}||a$ and $\lambda = \lambda_{21} - \lambda_{22}$ for $\mathbf{k}||c$) depends on orientation of the Néel vector and attains its maximal values at $\mu_0 H \propto 3$ T. Further, hybridization between the magnons and phonons reaches a maximum for wave-vectors $k \approx k_0$, for which the group velocity of magnons, and the velocity of longitudinal phonons coincide, $v = d\omega_1(k_0; H)/dk \approx s_l$. Such hybridization allows for spin transport by phonons and gives possibly rise to a larger relaxation time τ .

Second, according to *ab-initio* calculations [T. Shen et al., *Optoelectronics and advanced materials – Rapid Communications* 10, 268-272 (2016).], the elastic modulus of YFeO₃ shows a strong anisotropy. We estimate the difference between $s_l(\mathbf{k}||a)$ and $s_l(\mathbf{k}||c)$ to be 20%. This also means a 20% difference between the velocity of hybridized magnons with $k \approx k_0$ propagating along a and c axis. Assuming that the spin signal is transported mainly by these particular magnons, we anticipate additional splitting of the magnon group velocities v_x and v_y .

So, while magnetoelastic coupling can play a role, we do not probe the impact of varying strain on the spin transport. However, we mention in the revised manuscript that this is of course of interest for a future study beyond the scope of the current work.

We have now revised the manuscript in response and added significant new information to the revised version.

Comment 2: Again, I concern the accuracy of the damping constant of YFeO₃ extracted from the experimental data. The authors present the damping constant of $(3 - 6) \times 10^{-6}$, which seems the lowest value reported so far, to my knowledge. After checking the raw data, I am not sure how reliable the value is. I suggest the authors should carefully analyze the data.

Our response: We thank the referee for raising this point. The damping constant for an antiferromagnetic material inherently accounts for the strong internal exchange coupling between the two sublattices. This field is 635 T for YFeO₃, which explains the low magnitude of the damping.

Our analysis was carried out in a very controlled manner, which we recall further below. As a result, we have been able to verify that the data we obtain are reproducible (Figs 1 to 3 below, also used as revised Figs 1(c) in the main text and Fig. S6 and S7 in the Supplementary materials, respectively). In addition, the data we obtain agree with the expected temperature-

dependence, i.e., a larger value of damping for higher temperatures, related to the increase in magnonic and phononic contributions that open up relaxation channels. Finally, we would like to mention that we have been particularly careful not to underestimate the value of damping and that our analysis provides an upper bound of damping. Therefore, we consider our analysis and interpretation solid and carefully performed.

We recall below in detail how we conducted our analysis.

We extract the average linewidth from the smallest peak-to-peak distances of the resonance spectra, which relate to the damping coefficient. These distances are indicated in the supplementary figures S6 to S9. The error bars are determined by the maximum deviation of the average value.

Because we use a wavelength in the same order as the dimension of the sample, at low fields there is an additional linewidth broadening and peak overlap due to the presence of magneto-static modes. This issue is addressed when using smallest peak-to-peak distances of the resonances as we do.

Resonance data are measured for the two cases discussed in the manuscript: $H \parallel a$ (easy axis) and $H \parallel c$ (intermediate axis). For $H \parallel c$, resonance data were measured several times, after removal and repositioning of the sample to test data reproducibility. The linewidth dependence is fitted using the theoretical model of antiferromagnetic resonance provided by Fink (Ref. 42 of the manuscript, please see Fig. 1(c) of main text and Fig. S5 of the supplementary), which yields a damping of $3.5 \pm 0.4 \times 10^{-6}$ for data set 1 and $3.4 \pm 0.3 \times 10^{-6}$ for data set 2 at 20 K and $6.2 \pm 0.3 \times 10^{-6}$ and $4.6 \times \pm 0.5 \times 10^{-6}$ at 150 K, respectively, for the field along the c-axis and $6 \pm 2 \times 10^{-6}$ at 20 K for the field along the a-axis. We would like to point out that this still corresponds to an upper bound for the magnetic damping.

Figure 1: Linewidth as a function of frequency for the configuration of H along c-axis at 20 K and 150 K. The open and closed symbols correspond to data taken separately, after removal and reintroduction of the sample, to test reproducibility. The solid and dashed lines are the theoretical fitting using the model from Fink (Ref. 42 of main text), which yields damping coefficients of $3.5 \pm 0.4 \times 10^{-6}$ and $3.4 \pm 0.3 \times 10^{-6}$ for the two sets of data at 20 K and $6.2 \pm 0.3 \times 10^{-6}$ and $4.6 \pm 0.5 \times 10^{-6}$ for 150 K. The extracted parameters from Fig. 1(b) of the main text, $\mu_0 H_E = 635$ T, $\mu_0 H_a = 0.19$ T, $\mu_0 H_b = 0.7$ T and $\mu_0 H_{DMI} = 12$ T, are used in this model.

Figure 2: Resonance peaks at different frequencies for a 1 mm thick YFeO_3 (1 mm)/ Pt (5 nm) sample for the configuration of H along c-axis (intermediate axis) at 20K, for two sets of data (columns a-e) and (columns f-i) taken separately after removal and reintroduction of the sample. The resonance FWHM linewidth is extracted by measuring the average peak-to-peak distance of the resonances, indicated by the red dashed lines.

Figure 3: Resonance peaks at different frequencies for a 1 mm thick YFeO_3 (1 mm) / Pt (5 nm) sample for the configuration of H along c-axis (intermediate axis) at 150K, for two sets of data (columns a-e) and (columns f-i) taken separately after removal and reintroduction of the sample. The resonance FWHM linewidth is extracted by measuring the average peak-to-peak distance of the resonances, indicated by the red dashed lines.

Reviewer #2 (Remarks to the Author):

In the revised manuscript, the authors have discussed the role of magnetoelastic coupling on anisotropy in magnon transport and the ferroelectric effect in YFeO₃. These results could be interesting for exploring the hybridization of antiferromagnetic magnons with acoustic phonons and ferroelectricity in antiferromagnetic materials. Therefore, I can recommend the manuscript for publication in Nature Communications after minor revisions.

(1) The fits using the model from Fink shown in Fig. 1c give extremely low damping constants. However, these fits do not agree well with the experimental data. I suggest the authors briefly describe the reasons for the discrepancy.

(2) In lines 165-166, the authors wrote "We note that the damping constant for an antiferromagnetic material inherently accounts for the strong internal exchange coupling between the two sub-lattices..." Please include the references.

Our response to Reviewer #2

Comment: In the revised manuscript, the authors have discussed the role of magnetoelastic coupling on anisotropy in magnon transport and the ferroelectric effect in YFeO_3 . These results could be interesting for exploring the hybridization of antiferromagnetic magnons with acoustic phonons and ferroelectricity in antiferromagnetic materials. Therefore, I can recommend the manuscript for publication in Nature Communications after minor revisions.

Our response: We thank the reviewer for acknowledging our efforts to improve the manuscript and for recommending the revised manuscript for publication in Nature Communications after taking care of two minor comments. We have adjusted the manuscript in response and our response to the comments is as follows.

Comment 1: The fits using the model from Fink shown in Fig. 1c give extremely low damping constants. However, these fits do not agree well with the experimental data. I suggest the authors briefly describe the reasons for the discrepancy.

Our response: We thank the reviewer for this comment. It is clear that the Fink model reproduces the gradual decrease of the linewidth with frequency, determined experimentally. The uncertainty in the experimental data and the resulting deviation are indicated by error bars, which are due to the fact that the resonance measurements show multiple peaks associated with the magnetostatic modes. This point has been clarified in the revised manuscript. The portion highlighted in yellow is the new text and the portion highlighted in green is the existing text supporting the answer.

Comment 2: In lines 165-166, the authors wrote “We note that the damping constant for an antiferromagnetic material inherently accounts for the strong internal exchange coupling between the two sub-lattices...” Please include the references.

Our response: The review article, which includes the equations of antiferromagnetic dynamics is the following:

Semenov, Y. G. & Kim, K. W. Modeling of Antiferromagnetic Dynamics: A Brief Review. *J IEEE Nanotechnology Magazine* **14**, 32-41 (2020).

The reference has been added in the revised manuscript.